# Evaluation of Abnormal Hypocotyl Growth of Mutant *Capsicum annuum* Plants

**Bánk Pápai** [1],*[ID], **Zsófia Kovács** [1][ID], **Kitti Andrea Tóth-Lencsés** [1], **Janka Bedő** [1], **Gábor Csilléry** [2], **Anikó Veres** [1] **and Antal Szőke** [1]

[1] Department of Genetics and Genomics, Institute of Genetics and Biotechnology, MATE Hungarian University of Agriculture and Life Sciences, Páter Károly Utca 1, 2100 Gödöllő, Hungary
[2] PepGen Kft., Bartók Béla Út 41, 1114 Budapest, Hungary
* Correspondence: papai.bank@uni-mate.hu; Tel.: +36-30-864-32-67

**Abstract:** Horticulture is a dynamically evolving and an ever-changing sector which needs new ideas, plant materials, and cultivating methods to produce more. Involving different mutants in breeding lines may lead to new opportunities to create new cultivating methods. *pcx* (procumbent plant) and *tti* (tortuosa internodi) *Capsicum annuum* mutant plants, which present abnormal stem growth, were investigated in various in vitro experiments. The *pcx* breeding line presents highly diverse hypocotyl growth even in the early phenophase, such as normally growing plants and the 'laying' habit. On the other hand, *tti* plants only present their elongated slender stem trait in a more mature phase. In our experiment of reorientation, we used one-sided illumination, where each of the phenotypes sensed and reacted to light, and only the *pcx* plants exhibited a negative gravitropic response. It was also the result that the *tti* plants sensed gravity, but the weak structure of the hypocotyls made them incapable of following its direction. Since the *pcx* plants were the only ones with an 'antigravitropic' growth, we used them to evaluate the time course they needed to adapt and follow the gravity vector after reorientation. The *pcx* plants sensing gravity adapted similarly to controls and started bending after 120 min, but those which presented as 'anti-gravitropic' did not respond even after 420 min.

**Keywords:** *Capsicum annuum*; gravitropism; phototropism; mutation; hypocotyl

## 1. Introduction

Higher plants can adapt and react to various environmental stimuli such as light, gravity and touch [1–3]. This capability to adapt made them able to leave the water and grow on the surface [4,5]. The phenomenon when a plant's growth is mediated by the gravitational force as a stimulus to direct the shoot, root and leaf positioning is called gravitropism. Proper plant posture is crucial for plants to reach sunlight efficiently to photosynthesize [6,7]. The ability to sense gravitational force is also essential to direct the root growth downwards into the soil to reach water and minerals which are vital elements of the plant's life cycle [8–10]. The gravitropic process is considerably complex; it consists of four consecutive steps which are gravity perception, signal formation in the gravity-sensing cell, intra- and intercellular signal transduction, and asymmetric cell elongation between the upper and lower sides of the responding organs [11–13].

Gravity perception starts with the gravity sensors in statocytes, which contain the amyloplasts-statoliths. These are specialized gravity-perceptive cells and since they are heavier than the surrounding cytoplasm, they fulfill a statolithic function to sediment in the direction of the gravitational vector to the bottom part of the cells under the influence of gravity [14–16]. In roots, they take place especially in the root cap, which covers the root apical meristem. In plant stems, the endodermis contains the statocytes in hypocotyls, epicotyls, inflorescences, flower stalks, peduncles, gynophores, and in leaf petioles, among others [17]. After the perception of the gravitational pull, the information is converted

from mechanical to biochemical signals through cellular organelles, such as membranes and cytoskeletons, which results an asymmetric auxin flow in plants [18–20]. The auxin hormone disposition influences the cell elongation on the side of the stem, root, etc., which results in bending in the direction of gravity [21–24]. Mutations responsible for any disfunction in the gravitropic process may lead to reversed gravitropism, which we call the "anti-gravitropic" phenotype here [25–27].

Various environmental stimuli justify the necessity of creating an environment capable of the gravitropic analysis of the plant material [28]. In vitro experiments make it possible to observe plant growth without any biotic or abiotic interfering factors with a defined culture medium and controlled environmental conditions [29]. In addition to these advantages, in vitro experiments provide us with more consistent plant material. They can be analyzed under sterile conditions and different plant organs can be observed separately [30]. Most gravitropic experiments use 90° reorientation for observation of the plants' response to gravitational force [31–35].

The Hungarian bell pepper has an important role in traditional Hungarian cuisine and food consumption [36–38]. The value of the bell pepper fruit is determined by the size, the color, the shape, and mostly the taste [39,40]. Because of its economic importance to Hungarian bell pepper fruits [41,42], it is necessary to create and improve new cultivating styles to produce a greater yield. Horticulture is an ever-changing sector which needs new ideas and innovations to compete with climate change and the growing population [43,44]. Different stem mutations seemed to be harmful mutations with no usable value in the past. Nowadays, modern plant breeders may use mutant traits to improve the cultivars and create new cultivating methods [45–47]. Among others, leaf and branch growth angles may partially affect the yield, which means this may make them a target for improvement in plants [48]. The evaluation of plant hypocotyl growth is important to check to see if mutant plants possess valuable traits which might be useful to apply into breeding lines. In this article, we use an in vitro culture methodology which was developed to observe and compare the hypoctyl growth of these mutants. We further sought to investigate if the abnormal growth may be explicable from the disfunction of the photo- and gravitropic mechanism of the hypocotyls.

## 2. Materials and Methods

### 2.1. Plant Materials

The study compared two breeding lines which develop their stems abnormally, and additionally, a commercially available processing variety as a control. The mutant pepper breeding line seeds were obtained from Hungarian pepper breeder Gábor Csilléry, PepGen Kft., Budapest, Hungary. The procumbent plant (*pcx*) and tortuosa internodi (*tti*) mutant (Figure 1) breeding lines were collected and maintained by self-pollination. For the following experiments, we used the seeds of the self-fertilizing breeding lines.

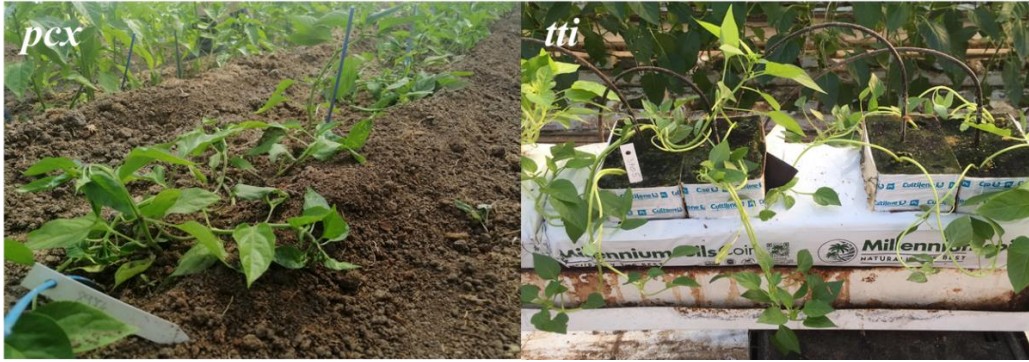

**Figure 1.** The *pcx* (**right**) and *tti* (**left**) plants grown in greenhouse in Szentes, Hungary.

The *pcx* phenotype was first observed in the F3 progeny of a spice-type bell pepper breeding line, whose fruits mature from dark green to red. The *pcx* plants showed diverse stem-growth types. The mutant trait was easily observable even in the early phenophase, as practically all the plants presenting the mutant trait laid low on the ground after germination. Some plantlets kept their growth habit, while some of them continued their development by turning the hypocotyls upwards later on. The laying mutants' development lagged behind compared to the normally growing plantlets from the same progeny. The plants which presented the laying trait only in their early development stage showed a strongly upwardly growing apical stem but laying offshoots. Even though the *pcx* plants presented those previously mentioned abnormal growth habits (Figure 2), the shoots did not wither and their structure is significantly rigid. Based on previous test results, it is hypothesized that a recessive gene is responsible for the mutant trait and there may be some other modifying genes intervening.

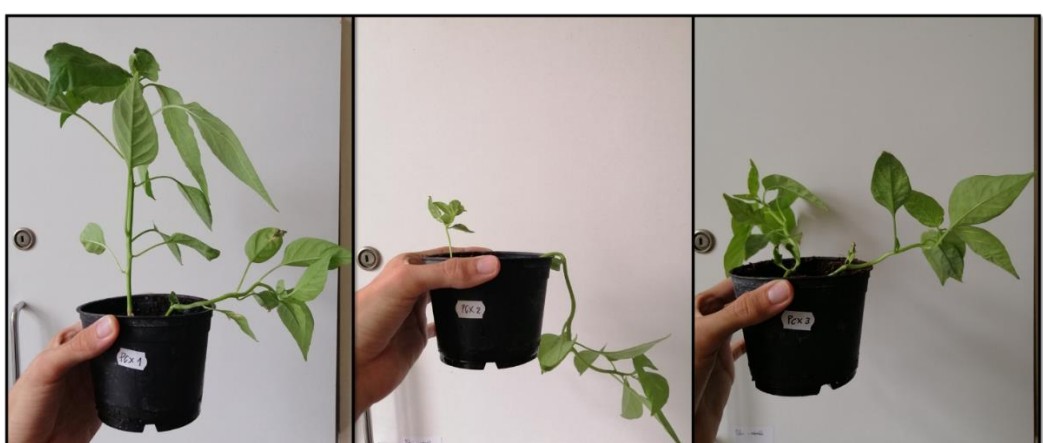

**Figure 2.** Various stem growth habit of the *pcx* plants.

The first *tti* phenotype was found in the F3 progeny of a breeding line, which have Cecei type fruits, which mature from white to red. The *tti* plants had elongated hypocotyl and they grew a long spiral slender stem; the internodes of the plants were much longer than usual. In the case of the *Capsicum annuum*, the normal internode length was usually between 5–8 cm, while the *tti* plants had 8–11 cm long internodes on the apical stem; in the case of the offshoots, they might have been even longer. The mutant trait was not easily observable in the early phenophase; the stem usually bent and spiralled after developing 3–4 leaves. The structure of stems was weak and it broke easily under pressure. Considering previous results, it is presupposed that for this mutant trait, only a single recessive gene is responsible. However, we do not know the exact genetic background of either of the mutant traits. We called the plant materials *pcx* and *tti* in this initial stage of selecting new varieties.

As a control, we used the Hungarian cultivar 'Fehérözön' (Royal Sluis Magrovet Kft., Kecskemét, Hungary).

### 2.2. Experimental Place

The experiment was carried out in the Cell and Tissue Culture Laboratory of the Hungarian University of Agriculture and Life Sciences, Institute of Genetics and Biotechnology, Department of Genetics and Genomics, Molecular Biology and Breeding Group.

### 2.3. Experimental Procedure

All the following steps were carried out under completely sterile conditions in a laminar airflow. The seeds of the *pcx*, *tti* and 'Fehérözön' were surface sterilized using 70% ethanol rinsed for 45 s followed by a 1% sodium hypochlorite solution for 20 min, with continuous shaking to effectively remove all kinds of pathogens. Then, they were rinsed with sterile distilled water three times to remove all the remaining sodium hypochlorite

from them. The seeds were then placed on sterile filter paper, dried, and placed in glass vessels containing MS medium supplemented with 3% saccharose and 0.4% plant agar. The pH of the plant medium was at an optimal level of between 5.6 and 5.8. Then, it was autoclaved at 121 °C at 15 psi pressure for 20 min. The seeds were germinated at 25 ± 1 °C for 16 h–8 h light–dark periods and at 5000 lux light intensity using a BINDER Model KBWF 240 Growth chamber (BINDER GmbH, Tuttlingen, Germany) and Osram BIOLUX T8 L 18W/965 G13 fluorescent lamps (ams-OSRAM AG, Premstaetten, Austria). Two- and three-week-old seedlings were documented to examine the hypocotyl growth. The hypocotyls' length and curvature (°) were measured. The three-week-old seedlinga were bent by 90° following the methodology by Grube [49] to check the gravitropic response (Figure 3). These plantlets were documented after 24 h and their hypocotyl curvature was measured (°). To evaluate the photo- and gravitropic response of the mutants and the control, their seeds were germinated on MS medium, following the previous methodology, using one-sided illumination and complete darkness. After these seedlings became three weeks old, they were reoriented by 90° and documented 24 h later for further analysis. To evaluate the time course needed for the plants to react and follow the gravitational pull, and then to reach their final posture, we followed the methodology by Fukaki [50]. Three-week-old seedlings were bent by 90 degrees, and then the plantlets were documented every hour. Statistical analyses were performed using Microsoft® Excel® LTSC MSO (16.0.14332.20438) 64-bit software, using the one-way analysis of variance (ANOVA) procedure.

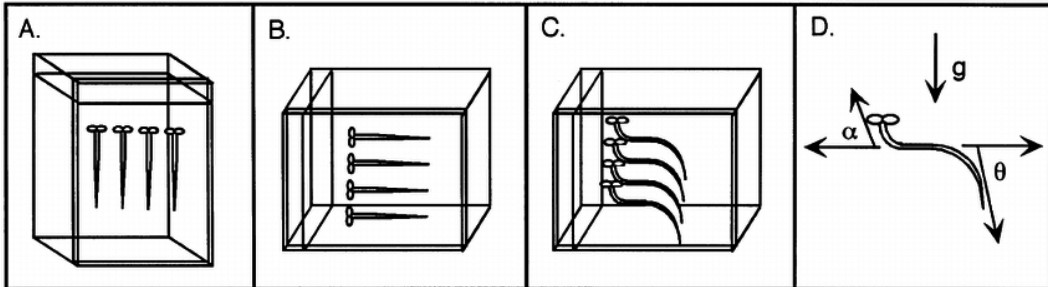

**Figure 3.** Method used to evaluate graviresponse in seedlings Straight seedlings were placed on solid plant media (**A**), which was then rotated 90° (**B**). Responding seedlings reoriented to resume vertical growth (**C**). Measurements taken included hypocotyl curvature. α, hypocotyl curvature; θ, root curvature; g, direction of the gravity vector. After 24 h, the hypocotyl curvatures were checked (°) (**D**) [49].

## 3. Results

### 3.1. Hypocotyl Growth of Two- and Three-Weeks Old Plantlets In Vitro Conditions

Our experiment focuses only on the photo- and gravitropic response and the abnormal bending of the hypocotyls of these *Capsicum annuum* seedlings. The bending of the hypocotyls seems to be independent of the root curvature, because in all cases, the roots followed the gravitational pull, growing downwards. Records of hypocotyl growth and bending in the direction of the gravity vector of the two- and three-week-old seedlings are shown in Figure 4.

Using in vitro methodology made it possible for us to prohibit any abiotic stress-factor to influence the plant growth. During our experiment, we observed that even one week of growing time can result in a much different hypocotyl angle. In the case of *pcx* plants, we perceived three different hypocotyl growths. There were plantlets which grew normally, those which had bent at first but started to grow upwards after a while, and those which stayed in their laying phase. In the case of *tti* plants, we observed an only slightly longer hypocotyl, but compared to the control plants, they presented a normal growth habit following the gravity vector. Plant height was measured in cm and the curvature was measured every 0.1 cm from the hypocotyl base, for 10 plants from each genotype. Then, we summarized the data in an Excel graph, and the results are shown in Figure 5.

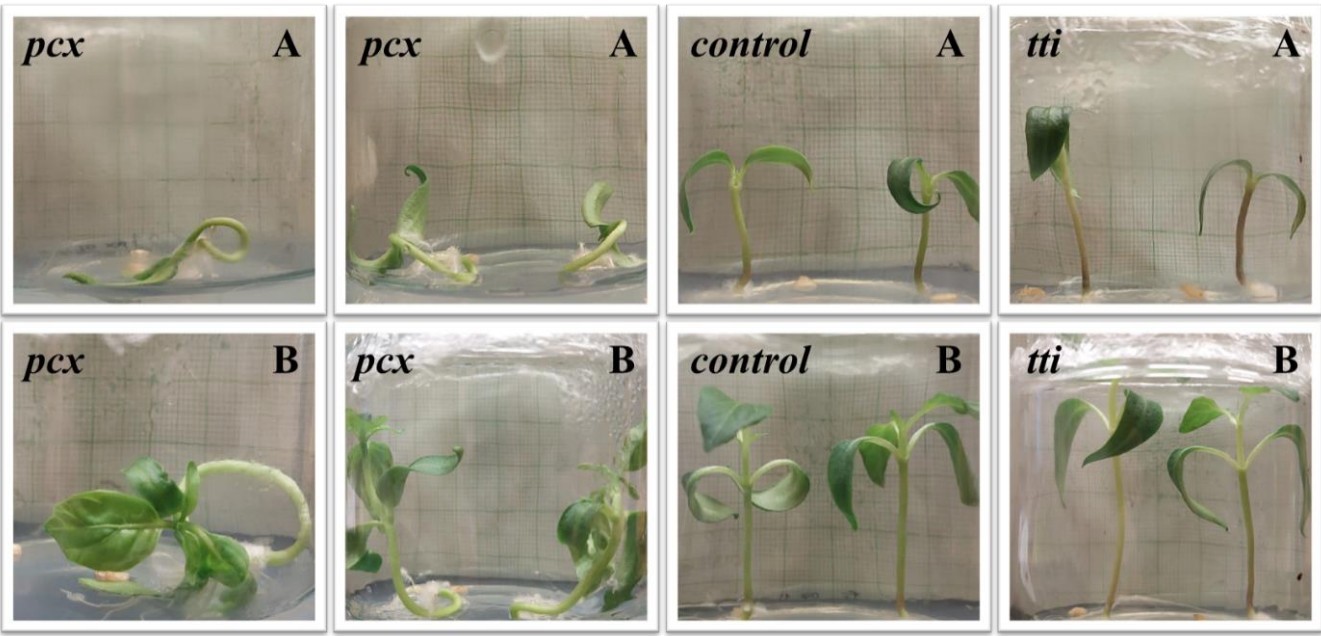

**Figure 4.** Two- (**A**) and three-week-old (**B**) in vitro seedlings of *pcx*, *tti* and control plants.

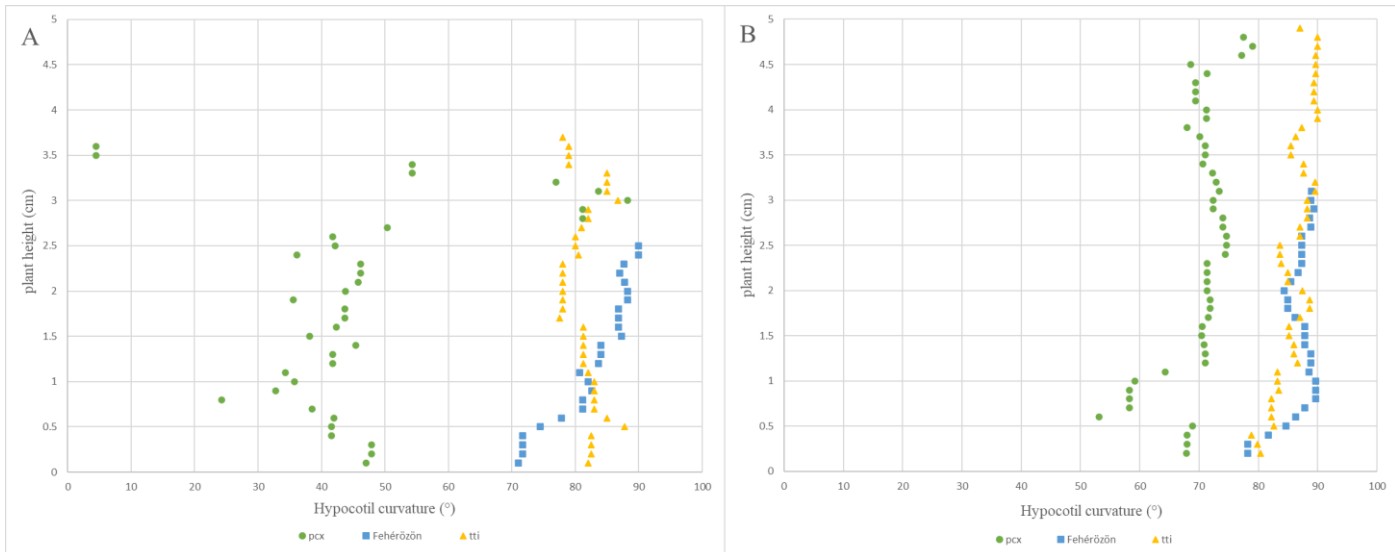

**Figure 5.** Curvature of the hypocotyl base of *pcx*, *tti* and control *Capsicum annuum* seedlings after being grown for two (**A**) and three weeks (**B**) under in vitro conditions. Plant height is measured in cm and the curvature was measured every 0.1 cm from the hypocotyl base. n = 30.

A one-way between-subject ANOVA was conducted to compare the hypocotyl curvature between the two- and three-week-old genotypes. There was a significant difference between the two-week-old genotypes in hypocotyl curvature at the $p < 0.05$ level for the three conditions [$F_{(2, 98)} = 117.1$, $p = 1.05008 \times 10^{-26}$]. There was also a significant difference between the three-week-old genotypes in hypocotyl curvature at the $p < 0.05$ level for the three conditions [$F_{(2, 125)} = 250.02$, $p = 2.05612\text{E}{-44}$].

After summarizing our data, the main finding of our study is the abnormal growth of the hypocotyl in the case of the *pcx* plantlets. The *tti* and the control plants' hypocotyl bending always converged to the almost 90 degrees, presenting a normal gravitropic response. In the case of our *pcx* mutants, there was a great difference between plants from the same progeny. After two weeks, we obtained a huge deviation in hypocotyl angles. After one more week, the same plantlets were measured. The three-week-old plants had

results closer to the 90 degree position. We explain it by what we mentioned before, that some of the plantlets had a laying position only in the early phenophase. Considering these results, we reoriented our glass vessels containing the in vitro plants by 90 degrees, and 24 h later, we checked the hypocotyl bending, as shown in Figure 6.

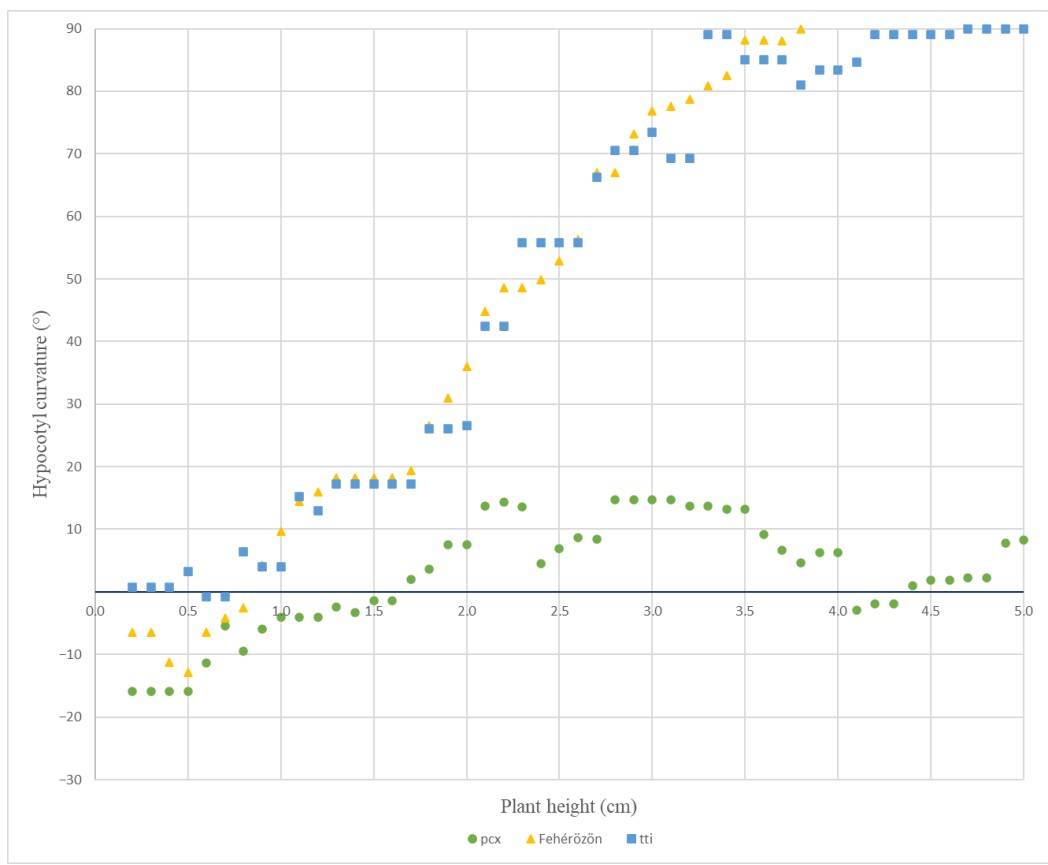

**Figure 6.** Curvature of the hypocotyls of *pcx*, *tti* and control *Capsicum annuum* three-week-old seedlings 24 h after reorientation by 90° under in vitro conditions. Plant length is measured in cm and the curvature was measured every 0.1 cm from the hypocotyl base. n = 30.

Considering this data, the only plantlets presenting an abnormal hypocotyl growth in response to gravity were the *pcx* plantlets. The *tti* and the control phenotypes all reached a point around 90 degrees, while the *pcx* plantlets presented hypocotyls differently. Considering their growth, our goal became checking the phototropic and gravitropic response of these phenotypes. A one-way between-subject ANOVA was conducted to compare the hypocotyl curvature between the different genotypes after reorientation. We identified a significant difference between the reorientated genotypes in hypocotyl curvature at the $p < 0.05$ level for the three conditions [$F(2, 137) = 41.24$, $p = 9.53938E{-}15$].

### 3.2. Photo- and Gravitropic Response of the Different Phenotypes

Since we know that the pathways for perceiving light and gravity intervene to determine plant growth [51], in our following experiment, we wanted to check how the different phenotypes reacted to different stimuli. Plants can adjust their growth considerably fast depending on the direction of light. During our experiment, in vitro seedlings received only one-sided illumination or no light at all (Figure 7).

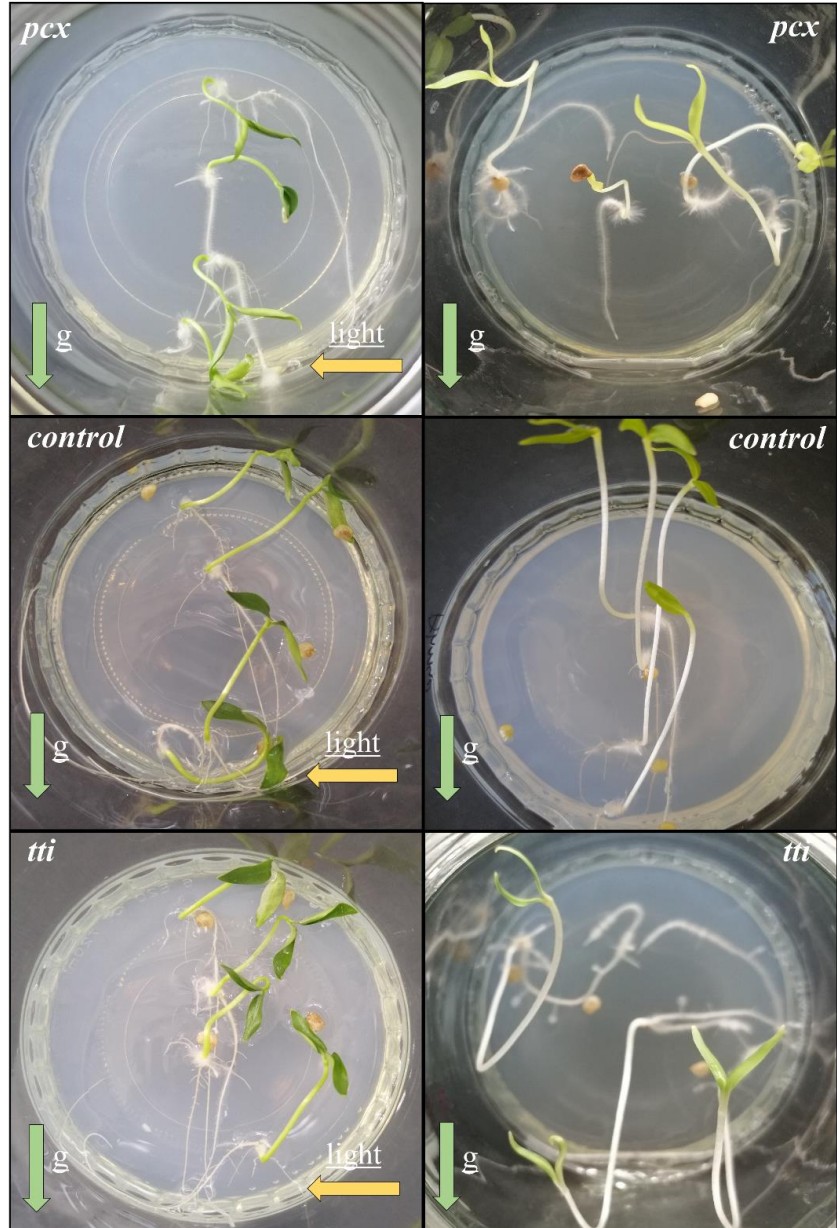

**Figure 7.** Photo- and gravitropic response of three-week-old *pcx, control* and *tti* plants reoriented by 90 degrees and documented after 24 h under in vitro conditions.

According to the experimental results, all the plants showed a very strong phototropic response, moving their cotyledons and bending the hypocotyls following the direction of the light. The control and *tti* plants presented a graviresponse; however, the *tti* plants bent their hypocotyls downwards, while moving the tips upwards in the end, suggesting that, despite their gravitropism, their weak structure and elongation made them unable to keep themselves upwards. The *pcx* mutants reacted to the illumination, moving their cotyledons in that direction to reach a better position for photosynthesis; however, the hypoctyls inclined downwards. In a completely dark environment, no longer having light as the leading stimulus resulted in random hypocotyl growth. As we observed herein, roots always follow the gravitropic stimulus and grow downwards so we conclude that the 'laying' trait is independent from the root growth. Since we observed that only the *pcx* plants presented a different plant posture after reorientation, in the following experiment, we only included the *pcx* mutant.

### 3.3. Time Course of Change in Curvatures of Hypocotyl Growth in pcx Plants

In the following experiment, we used three-week-old plantlets to evaluate the time needed for the plantlets to respond and adapt to the gravity stimulus after a 90-degree reorientation. Plants were documented every hour. The results are shown in Figure 8.

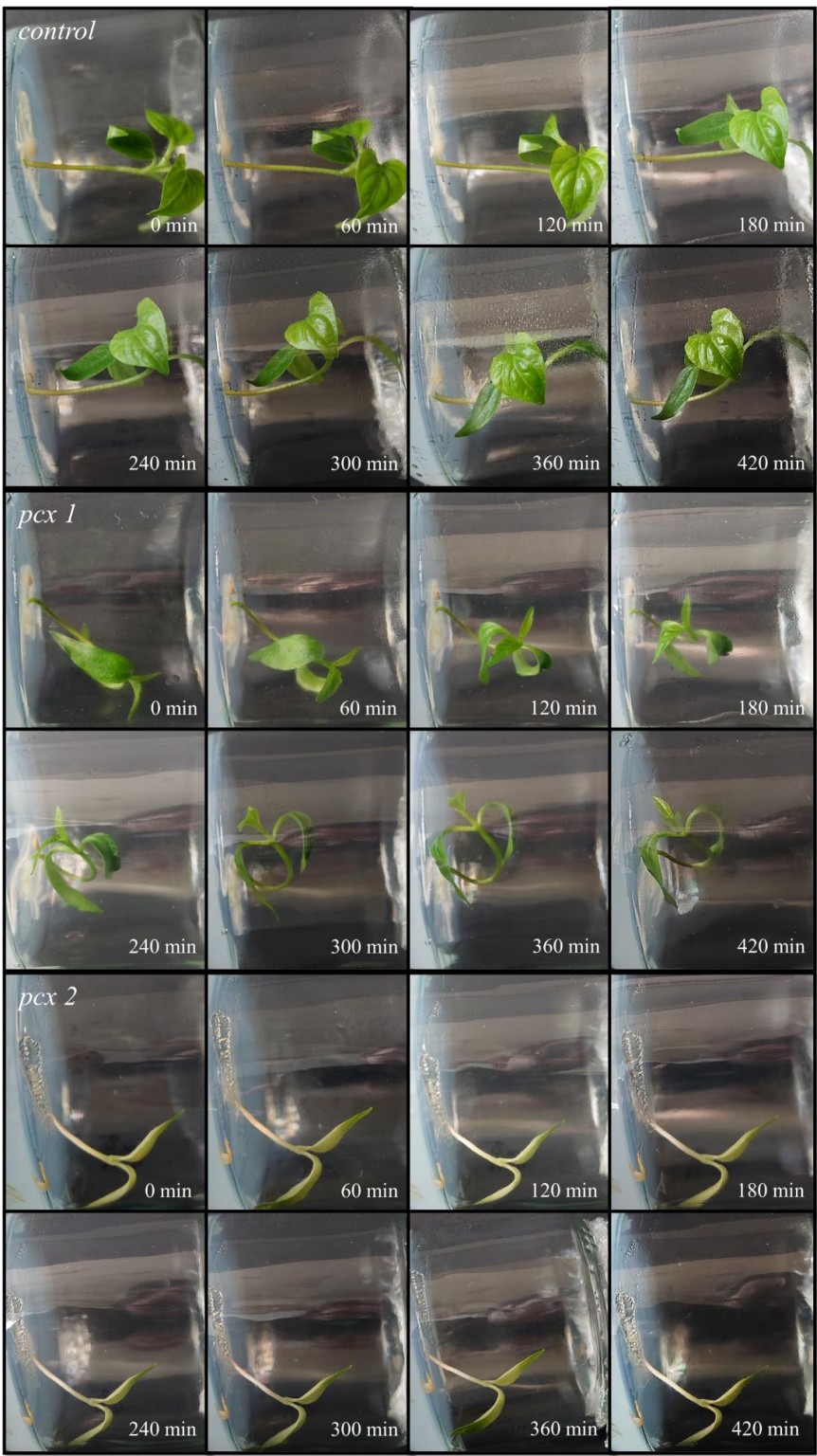

**Figure 8.** Gravitropic responses of the hypocotyls of three-week-old seedlings, documented every hour after a horizontal reorientation in the case of control and *pcx* plants.

In summary, we concluded that some *pcx* plants which had the ability to sense gravity reacted to a gravitational pull considerably fast. They had a slight bending even after 120 min, similar to controls. They reached the 90° hypocotyl bending similarly to the controls, while those *pcx* plants which had no graviresponse in the beginning did not present any hypocotyl bending even after 420 min. A one-way between-subject ANOVA was conducted to compare the time course needed for the hypocotyl bending after reorientation. Our result showed a significant difference between the genotypes at the $p < 0.05$ level for the three conditions [$F(10,88) = 12.8$, $p = 1.76776E-13$]. Since only the *pcx* plants present this phenotype, we suspect that they have some kind of mutation which affects a part of a gravitropic process, resulting in an 'anti-gravitropic' phenotype. The results of the experiment are summarized in Figure 9.

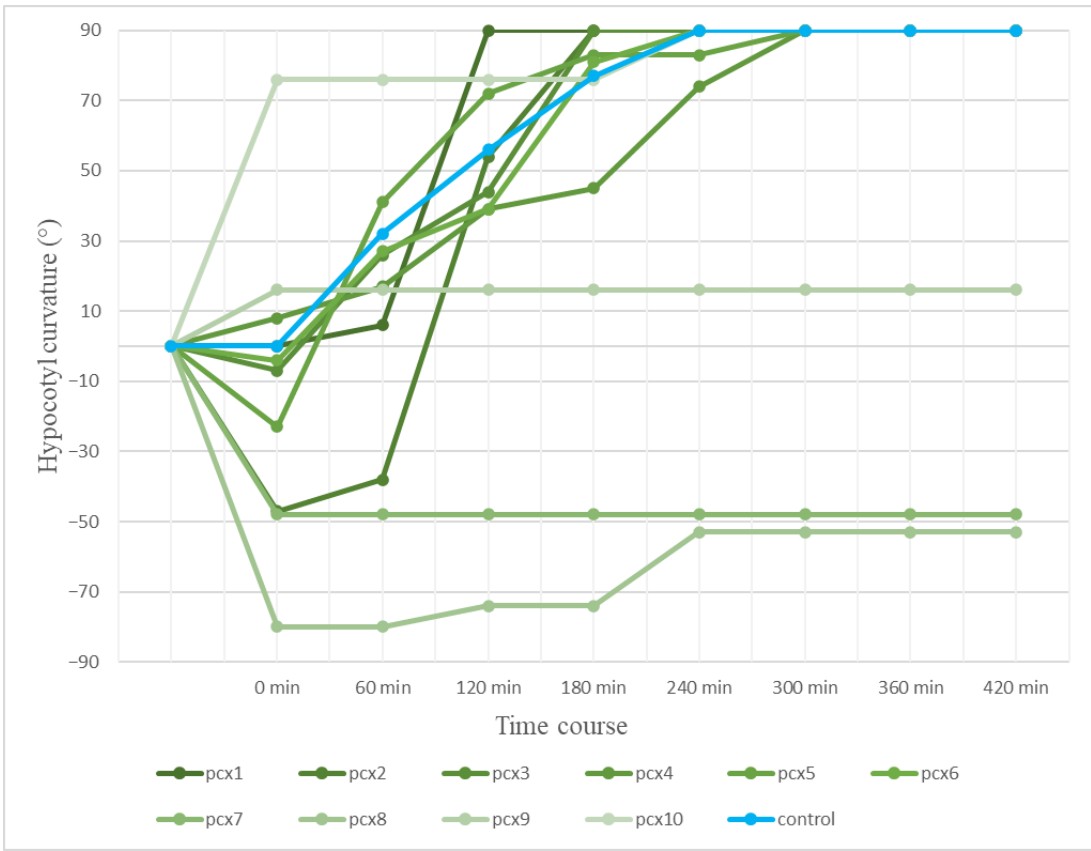

**Figure 9.** Time course of the gravitropic response of the hypocotyls of *pcx* and control *Capsicum annuum* three-week-old seedlings reoriented by 90°, documented every hour under in vitro conditions. Hypocotyl curvature was measured in °. n = 11.

## 4. Discussion

In this study, we included the *pcx* and the *tti* mutant breeding lines in our experiments to compare their early growth under in vitro conditions and how they reacted to different abiotic stimuli, such as light and gravity. We found that in the early phenophases, only the *pcx* mutants presented an abnormal hypocotyl development, which resulted in bending after weeks. However, some of the plantlets reached their proper hypocotyl posture when we measured their bending curvature after one more week of development. For us, it was previously known that these plants in the same progeny are very diverse, presenting different growth habits, which is easily observable in a very early phenophase, suggesting that there might be a developmental problem in the case of hypocotyls; furthermore, it can be one effective method for the selection of the plants.

Researchers have discovered various mutants which show similar phenotypes in various plants, such as *Arabidopsis*, maize, rice and barley [52–55]. Although these plants

have a bigger tilling angle, only the LAZY1 maize mutant presents a huge similarity to the *pcx* plant. The results of checking the early growth of the *tti* plants presented a longer hypocotyl in the early developmental stage, but they did not exhibit inclination in that early phase. Compared to what was previously known about the mutant breeding line, we obtained the same result, that whenever the *tti* plants reached a specific height, they started to bend their posture. We checked how the hypocotyls of these mutants reacted to abiotic stimuli, such as light and gravity. Our study focused only on the hypocotyl bending, since in every case the roots always reacted and grew in the direction of the gravitational pull, even after any kind of reorientation. Considering the photo- and gravitropic responses of our mutants compared to a normally growing control, we made observations that all the mutants and the control sensed and reacted to light, moving the cotyledons and bending the hypocotyls in its direction. In a completely dark environment, the hypocotyls of the *tti* plants inclined, but turned upwards in the end, proving that they possess the gravity-sensing skill. However, we suspect that the exceeding elongation and the weak structure of the hypocotyl made them unable to hold themselves completely upwards. These mutants share similarities with other elongated-internode mutations, which were observed in tomato, rape, pea and barley [56–59]. The most obvious difference was discovered in the case of the *pcx* plants, since these were the only mutants laying downwards in every case. These results suggested that the *pcx* plants had an abnormal gravitropic response, since we know they grow a rigid stem, which made them able to hold themselves upwards. Since we know that they reacted to illumination, we continued focusing only on the graviresponse by conducting an experiment to check the time course needed to react and adapt their posture after reorientation. The results presented that some of the *pcx* plants had a similar gravitropic bending to the controls, presenting a slight incline even 120 min after the 90-degree reorientation. At 300 min, all the plants which presented any graviresponse reached close to the 90-degree posture; however, other *pcx* mutants laying down presented no response to gravity at all. Considering how these phenotypes reacted to the gravitropic stimuli and what we know about the mutant breeding line, we suspect that they may possess some unidentified mutations intervening in the gravitropic process. Various mutations can lead to an 'antigravtiropic' phenotype, which were summarized by Kawamoto and Morita [60].

## 5. Conclusions

In conclusion, this is the first report on the development of these mutants exhibiting these abnormal stem growth habits. These mutants might be useful resources in plant breeding. Overall, we found that the *tti* mutants respond to light and gravity as well; therefore, we suspect that their bending occurs because of the weak elongated structure of the stem. The *pcx* mutants are very diverse. During our study, we found that the laying plants presumably did not respond to gravitational stimulus at all after reorientation. Our previously mentioned results suggest that further long-term experiments are needed to understand the background of these abnormal stem growth habits. We also suggest analyzing different physiological and genetic factors, which are crucial to gravity sensing and responding in plants, and also checking the key factors for reaching a proper stem posture. Furthermore, understanding the elongated-internode trait as well might prove useful in the future. Using mutant plant materials in greenhouse vegetable production in different cultivating styles such as vertical farming would be also recommended to check the expected yield if the stem is grown differently.

**Author Contributions:** Conceptualization, G.C.; methodology, B.P. and Z.K.; validation, A.V. and A.S.; formal analysis, K.A.T.-L. and J.B.; investigation, B.P.; data curation, K.A.T.-L. and J.B.; writing—original draft preparation, B.P.; writing—review and editing, A.V. and A.S.; visualization, B.P.; supervision, G.C., A.V. and A.S. All authors have read and agreed to the published version of the manuscript.

**Funding:** Supported by the ÚNKP-22-3-II-MATE/25. new national excellence program of the ministry for culture and innovation from the source of the national research, development and innovation fund.

**Institutional Review Board Statement:** Not applicable.

**Data Availability Statement:** The data presented in this study are available on request from the corresponding author. Informed consent was obtained from all subjects involved in the study.

**Conflicts of Interest:** The authors declare no conflict of interest.

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
