# Peer review of "Evaluation of Abnormal Hypocotyl Growth of Mutant Capsicum annuum Plants"

_agriculture, doi:10.3390/agriculture13020481_

Round 1

Reviewer 1 Report

The authors investigated the hypocotyl morphology of two pepper mutant plants (pcx and tti) and their responses to one side illumination. The results are quite interesting. These two mutant materials may be useful for vertical farming. However, the writing of this manuscript should be improved. The authors have only observed and determined the morphology parameters. The physiological or molecular mechanisms are not studied. If possible, please add some data of physiological or molecular aspects. 

Author Response

Dear reviewer,

Thank you kindly for your time to check through the submitted manuscript and thank you for your suggestions as well. During our experiments we only checked the morphology parameters of these bell pepper mutant plants until now, however we plan to do a molecular analysis regarding the genes playing roles in the gravitropic process and we also have further plans to check the hormonal levels in the stems of these mutants used in various cultivating methods. To improve the manuscript we rewrote the Materials and Methods, the Results, the Discussion and the Conclusions parts. We thank you for you valuable comments and your time taken to help us improve our manuscript.

Reviewer 2 Report

Avoid unwanted spacing between abstract and keywords

Remove excess spacing after 2.1

Remove excess spacing from the line 131-141

Avoid excessive spacing between the paragraphs

Explain the discussion elaborately

Why particular time period was taken as a tool for evaluation in the experiment

Arrange the references in ascending order

Brief the legend of figure no 7 and resize the figure in uniform order

Author Response

Dear reviewer,

Thank you for your valuable time to read through the submitted manuscript and we are grateful for your useful suggestions as well. We removed the suggested excess spacing and between the paragraphs as well. Following your kind suggestion we resized and formed the figure number 7. We also modified the legend under that. We also reordered the references in ascending order. Thank you for your question about time period we chose for our research. During the experiment where we checked the time period needed for the plants to react to the gravity force we documented the seedlings hourly similarly to other experiments related to gravitropism. In all cases after reorientation 300 minutes were enough for all the plantlets to reach their final posture. To prove they did not move any further we observed them for another 120 minutes. Thank you for your comments to improve the quality of the manuscript, we rewrote the methodology, the experimental results, the discussion and the conclusion parts to meet the requirements.  

Reviewer 3 Report

This study provided two interesting pcx and tti pepper mutants, and described their stem phenotypes. I have a few questions about this study.  What is the basis for defining pcx and tti? If it is a gene name, the corresponding gene number needs to be added. The inconsistent phenotypes among different pcx plants indicate that the mutant may not be homozygous. I suggest using tissue culture seedlings to ensure similar genetic background. In addition, there is no additional evidence to prove that the phenotype of pcx mutant is caused by gravity perception rather than abnormal development of vascular tissue?

Author Response

Dear reviewer,

Thank you for your kind help to read through the submitted manuscript and thank you for your useful suggestions as well. The mutant genotypes are collected and maintained by the pepper breeder Gábor Csilléry. He named these phenotypes ’procumbent plant’ since their abnormal growth and ’tortuosa internodi’ for the long, slightly spiraling stem. The ’pcx’ and the ’tti’ are their abbreviations. They are written with small letters, since considering the results he assumes they may be recessive traits but the exact genetic backgroud is still unknown. We elaborated that part in our Materials and Methods chapter more. Our further plans to do molecular genetic experiments to check the genes which they identified in similar mutants. We are grateful for your comment on the plants may not be homozygous, the plant breeder tries to self-pollinate them and selecting the progeny exhibiting the procumbent trait. We will try to use an adequate protocol using tissue culture to micropropagate the plants to get uniform individuals. There might be various reasons behind the procument trait, however during our experiments we only measured morphological properties. As we mentioned in the Plant Material parts the pcx plants have a very rigid stem which suggested it might be a problem in the gravitropic process instead of possessing an abnormally lignified tissue, however taking your comment as a suggestion might worth planing more experiments about observing the vascular tissue of the stems. Thank you for your comments to improve the quality of the manuscript, we rewrote the methodology part with more detailed informations about the plants. We also corrected the results, the discussion and the conclusion parts as well to meet the requirements. Thank your for your contribution to make this article a better work.